# CROSS-PROTEIN WASSERSTEIN TRANSFORMER FOR PROTEIN-PROTEIN INTERACTIONS

## ABSTRACT

Previous studies reveal intimate relationships between the structure and function of proteins. Motivated by this, for protein-protein interactions (PPIs), we hypothesize that cross-protein structural correspondence, including both global correlation and local co-occurrence, poses a great influence. Accordingly, a novel deep learning framework named Cross-Protein Wasserstein Transformer (CPWT) is proposed to predict PPI sites through fine-grained cross-graph structural modeling. Considering the irregular architecture of acid sequences, for a pair of proteins, graphs are constructed to describe them. Then, a core Cross-Graph Transformer (CGT) module of two branches (e.g. ligand and receptor branches) is proposed for cross-protein structural modeling. Specifically, in this module, Wasserstein affinity across graphs is calculated through cross-graph query (i.e. ligand (query) - receptor (key) or the converse), based on which the multi-head attention is derived to adaptively mine fine-grained cues of PPI sites. By stacking CGT modules, the two branches in CGT are co-evolved in a deep architecture during forward inference, hence being powerful and advantageous in cross-protein structural representation and fine-grained learning. We verify the effectiveness of our CPWT framework by conducting comprehensive experiments on multiple PPI datasets, and further visualize the learned fine-grained saliencies for intuitive understanding.

## 1 INTRODUCTION

Proteins are chains of amino acids, and physical interaction between proteins is essential to life. The protein-protein interaction (PPI) (Figure 1 shows an example) determines molecular and cellular mechanisms, and thus plays a crucial role in biological processes including the gene expression, proliferation of cells, etc. Moreover, as proteins are predominant drug targets, characterizing PPIs at the fine-grained level, e.g. identifying protein interaction sites for PPIs, would provide significant insight into biological mechanisms, and has important application in drug design, disease treatment (Ryan & Matthews, 2005), and target discovery. For this reason, the research on PPI prediction has drawn increasing attention.

In the early stage, sundry experimental assays have been widely applied to PPI identification, such as nuclear magnetic resonance(NMR) (Wuthrich, 1989), X-ray crystallography (Svergun et al., 2001) and high-throughput screening methods (Song et al., 2011). However, identifying the binding sites based on these methods is often time-consuming and resource-expensive. Then, with more and more protein structure data available (Berman et al., 2000), computational methods are developed to predict protein-protein interaction sites, which can be divided into two categories, i.e. protein-protein docking and data-driven methods. For the protein-protein docking (Porter et al., 2019; Halperin et al., 2002), the fundamental principle is the steric complementarity at protein-protein interfaces. However, it suffers from the tremendous search space and requirement of expert-defined scoring functions in the searching and scoring processes for predicting complex structures.

For data-driven methods, a number of machine learning algorithms with shallow structures are first proposed for PPIs in the early stage. Generally, they can be divided into three categories, i.e. sequence-based methods (Murakami & Mizuguchi, 2010; Zhang et al., 2019; Jurtz et al., 2017; Haberal & Oğul, 2017; Zheng et al., 2018), structure-based methods (Du et al., 2016; Bradford & Westhead, 2005; Neuvirth et al., 2004) and those ones based on mixed information (Afsar Minhas et al., 2014; Li et al., 2012; Northey et al., 2018; Porollo & Meller, 2007). Then, inspired by the success of deep learning in vision tasks, deep neural networks are employed for protein science, e.g. the success of DeepMind's Alphafold2 (Jumper et al., 2021) for structure prediction. Specifically for PPIs, to utilize the powerful structure modeling ability, recent works (Bryant et al., 2022; Gao et al., 2022; Evans et al., 2021) including AF2Complex (Gao et al., 2022) and AlphaFold-Multimer (Evans et al., 2021) propose to

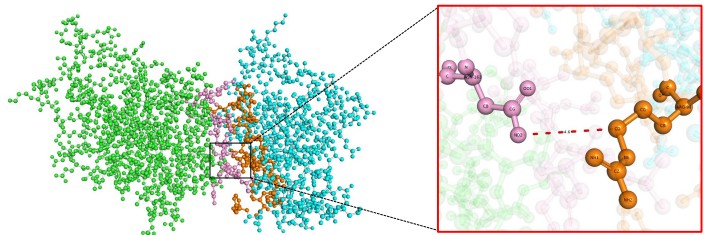

Figure 1: Ball and stick view of the protein complex named 2HRK, bound from a receptor protein (green) and a ligand protein (cyan). The interactive residues are marked as pink in the receptor and orange in the ligand. The zoomed-up area in the red box shows detailed interactions (red dotted lines) of amino acids between asparagine in the receptor (pink) and arginine in the ligand (orange).

predict the structure of multimeric protein complexes through the Alphafold model (Jumper et al., 2021), based on which interactions between protein sequences can be further identified. Moreover, multiple deep learning methods (Amidi et al., 2018; Torng & Altman, 2017; Jiménez et al., 2017; Skalic et al., 2019) attempt to map irregular atomic coordinates to regular representations of 3D grids, or just to 2D images which are then fed into Convolutional Neural Networks(CNN) (Gainza et al., 2020). Recently, graph neural networks (GNNs) (Fout et al., 2017; Liu et al., 2020; Morehead et al., 2022) are applied to deal with PPI prediction and achieve state-of-the-art performance.

Considerable progress has been made in PPI prediction, especially by GNNs. However, considering the intimate structure-function relationship, two main issues are remaining to be solved: (1) insufficient cross-protein structural modeling: the often used post-fusion of vectorized protein features fails to well characterize the cross-protein structural correlation; (2) unsophisticated fine-grained learning for PPI sites: interactive pairs of sites always take a pretty small proportion of residues (as shown in Figure 1) in proteins so that fine-grained modeling is rather necessary for PPI prediction, which is not adequately investigated in the previous study.

To tackle these issues, we propose a deep learning framework named Cross-Protein Wasserstein Transformer (CPWT) for PPI prediction. For a given pair of proteins, e.g. ligand and receptor, the target is to detect those interactive pairs of sites. Considering the irregular architecture of proteins and intrinsic relations among residues, graphs are constructed to describe proteins with residues as nodes and spatial relationship for defining edges, then fed to corresponding graph encoders for feature learning. Furthermore, a core Cross-Graph Transformer (CGT) module, consisting of two branches (e.g. ligand and receptor branches), is proposed to model structural correlation across proteins with fine-grained learning on PPI sites. In this process, cross-graph modeling is required, which is rather non-trivial. Theoretically, as irregular graphs don't lie in the Euclidean space, the common metric like cosine similarity, cannot well describe the cross-graph structural relationship. To address this issue, the Wasserstein metric is specifically introduced, based on which cross-protein query operations (i.e. ligand (query) - receptor (key) or the converse) are first conducted in the CGT by calculating Wasserstein affinities across graphs. Furthermore, multi-head attention is derived based on Wasserstein affinities to adaptively highlight salient pairs of sites and update the Transformer values.With the optimal transport principle of Waserstein affinity, the CGT is advantagous in characterizing two irregular (cross-graph) point/residue sets and exploring both global and local cross-graph structural information. Moreover, the CGT can be stacked in multiple layers so that its two branches can be effectively co-evolved in a deep architecture, hence being powerful in cross-protein structural expression and advantageous in fine-grained learning. We verify the effectiveness of our CPWT framework by conducting comprehensive experiments on PPI datasets, then visualize the fine-grained saliencies and compare them with the ground truth interaction for intuitive understanding.

The contributions are summarized as follows: (1) We propose a new Cross-Protein Wasserstein Transformer framework to promote the PPI prediction from the perspective of sophisticated cross-protein structural modeling based on Wasserstein affinities; (2) We propose a novel CGT module in which the multi-head attention is derived to mine fine-grained cues of PPI sites. Moreover, this module can be stacked into a deep architecture with two branches co-evolved, which are powerful

in cross-protein structural expression with sophisticated fine-grained learning; (3) We report the state-of-the-art performance on multiple PPI datasets with comprehensive experiments.

## 2 RELATED WORK

The PPI prediction problem has two flavors (Murakami & Mizuguchi, 2010; Zhang et al., 2019; Jurtz et al., 2017; Haberal & Oğul, 2017; Zheng et al., 2018; Afsar Minhas et al., 2014; Li et al., 2012; Northey et al., 2018; Porollo & Meller, 2007): the partner-independent prediction and the partner-specific one. The former is to predict whether a single residue of a protein could interact with residues from any other protein, while the latter is to detect interaction between residues from two proteins. Here, we focus on those works of partner-specific prediction, which is more relevant to our task. Generally, they can be roughly divided into two categories named surface geometry-based and graph-based methods, respectively.

**Surface geometry-based methods.** In order to encode and exploit surface geometry and interface complementarity in a deep learning framework, several methods have been proposed (Townshend et al., 2019; Behler & Parrinello, 2007; Sverrisson et al., 2021; Gainza et al., 2020; Sverrisson et al., 2021). Townshend *et al.* (Townshend et al., 2019) employs voxelization method which voxelizes local atomic environments or "surface patches", surrounding each of them, and then applies a 3D convolutional neural network to extract its latent features. Sverrisson *et al.* (Sverrisson et al., 2021) operates directly on the large set of atoms that compose the protein, generates a point cloud representation for the protein surface, learns task-specific geometric and chemical features on the surface point cloud and finally applies a new convolutional operator that approximates geodesic coordinates in the tangent space. Dai *et al.* (Dai & Bailey-Kellogg, 2021) constructs pairs of point clouds encoding the structures of two partner proteins, in order to predict their structural regions mediating interaction. It extracts local surface features and global protein features and then concatenated them to predict the interface regions on both interacting protein.

**Graph-based methods.** As one of the geometric deep learning methods, Graph Neural Networks (GNN) have been enthusiastically sought and have been proposed for protein interface region prediction (Fout et al., 2017; Duvenaud et al., 2015; Liu et al., 2020; Schütt et al., 2017; Pittala & Bailey-Kellogg, 2020). Font *et al.* (Fout et al., 2017) represents the protein as graph, which amino acids as nodes and affinities between nodes as edges. Then it employs GNN to learn node features, and classify each amino acid pair by a classifier. For better incorporating more informations. Liu *et al.* (Liu et al., 2020) proposes high-order interactions for protein interface. Same as (Fout et al., 2017), it learns node features by GNN. It proposes the sequential modeling method to incorporate the sequential information. Then it incorporates high-order pairwise interactions to generate a 3D tensor containing different pairwise interactions, and employs Convolutional Neural Networks(CNNs) to perform 2D dense predictions. Pittala *et al.* (Pittala & Bailey-Kellogg, 2020) combines the advantages of GNN and attention mechanisms in an integrated model for predicting both epitopes and paratopes. The former aggregate properties across local regions in a protein and the latter explicitly encode the context of the partner. Recently, Morehead *et al.* (Morehead et al., 2022) proposes the geometric transformer for rotation-invariant protein interface contact prediction. Moreover, the PepNN (Abdin et al., 2022) framework is proposed for protein-peptide interaction, where the reciprocal attention is employed for cross-graph measurement.

Different from all previous works, we propose a new CPWT framework from the perspective of cross-protein structural modeling. Specifically, we propose a novel CGT module consisting of two branches with multi-head attention based on the Wasserstein affinity, which is advantageous in cross-protein structural expression and fine-grained learning on PPI sites.

## 3 THE PROPOSED FRAMEWORK

### 3.1 OVERVIEW

The architecture of our CPWT framework with a one-layer CGT is shown in Figure 2, which takes a pair of ligand and receptor proteins as the input. Generally, the CPWT framework consists of three main learning processes including graph encoding, cross-graph transform, and PPI site prediction. Considering the irregular structure of proteins, graphs are constructed to describe them by taking residues as nodes with corresponding edges defined according to the spatial relationship. To learn robust features from these graphs, graph encoders such as GNNs are then applied. Furthermore, these encoded graphs are fed into the specifically designed CGT module to characterize structural

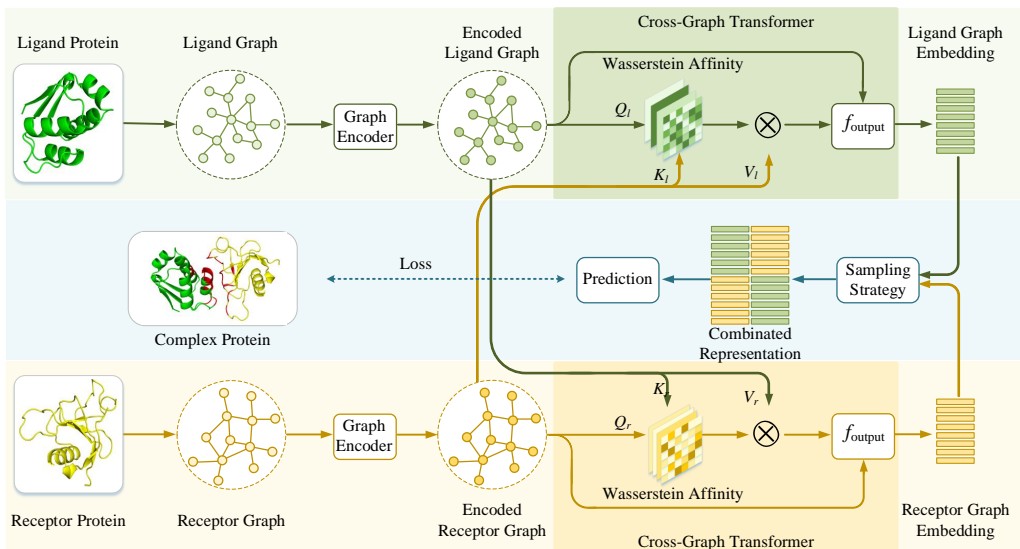

Figure 2: The architecture of our CPWT framework. Three main learning processes are involved in the CPWT framework, including graph encoding, cross-Wasserstein transformer, and prediction. The input pairs of receptor and ligand proteins are first modeled as graphs, which takes residues in amino acid sequences as nodes with edges existed in the k-nearest neighbors. Then, multi-layer GNN can be stacked as the encoder for robust feature learning. The encoded graphs are both fed into the CGT module for further fine-grained cross-protein learning. In this process, multiple CGT layers can be stacked into a deep architecture (only one layer shown here) with the two branches in CGT co-evolved. More details could be found in Section 3. Finally, the learned features of residues from two input proteins are combined for PPI site prediction.

correlation across two proteins and meantime achieve fine-grained learning. As shown in Figure 2, the CGT module consists of two branches that are mutually interacted and thus co-evolved. For each branch, two main operations, i.e. cross-graph query and multi-head attention, are involved. Specifically, the cross-graph query calculates the Wasserstein affinity across proteins where the sinkhorn algorithm is applied for cross-graph structural optimization. Based on query-resulted Wasserstein affinities, the multi-head attention is derived to update CGT values so as to adaptively detect salient pairs of sites. Finally, the learned features of residues in ligand and receptor proteins are combined to fulfill the prediction of those PPI sites. In the following parts, we descibe the learning processes in detail.

### 3.2 GRAPH ENCODING

For an input pair of ligand and receptor proteins, we first construct graphs for them which are denoted as $\mathcal{G}_l = \{(\mathcal{V}^l, \mathcal{E}^l) | \mathbf{x}_i^l \in \mathcal{V}^l, \mathbf{e}_{ij}^l \in \mathcal{E}^l\}$ and $\mathcal{G}_r = \{(\mathcal{V}^r, \mathcal{E}^r) | \mathbf{x}_i^r \in \mathcal{V}^r, \mathbf{e}_{ij}^r \in \mathcal{E}^r\}$, respectively. Here, $\mathcal{V}^l$ and $\mathcal{V}^r$ are the node sets of ligand and receptor graphs, while $\mathcal{E}^l$ and $\mathcal{E}^r$ are the edge sets. $\mathbf{x}_i^l$ denotes the feature vector of the $i$-th node while $\mathbf{e}_{ij}^l$ denotes the relationship (edge) feature between the $i$-th and $j$-th nodes in the ligand graph.

We adopt GNNs to learn nodes' representation based on its local neighbors. Considering that neighbours always contribute differently to the center node, different weights are distributed to them according to their distances from the center node, with larger weights for closer neighbors. Taking a node $\mathbf{x}_i^l$ in the ligand graph as an example, the graph encoding process is as follows:

$$\mathbf{h}_j^l = \sigma(\mathbf{W}_1^l \mathbf{x}_j^l + \mathbf{W}_2^l \mathbf{e}_{ij}^l + \mathbf{b}^l), \quad \alpha_j = softmax(\mathbf{h}_j^l \mathbf{p}^l), \tag{1}$$

$$\mathbf{z}_i^l = \sigma(\mathbf{W}_3^l \mathbf{x}_i + \frac{1}{|\mathcal{N}_i^l|} \sum_j \alpha_j \mathbf{h}_j^l), \tag{2}$$

where $j \in \mathcal{N}_i^l$ is neighbour set of the $i$-th node and $|\mathcal{N}_i^l|$ denotes neighbour node number, $\mathbf{W}_1^l, \mathbf{W}_2^l, \mathbf{W}_3^l$ are learnable projection matrices, $\mathbf{b}^l$ is the learnable bias, and $\mathbf{p}^l$ is an optimizable projection vector. $\sigma(\cdot)$ is a non-linear activation function, e.g. Relu, which results in the encoded feature denoted as $\mathbf{z}_i^l$.

### 3.3 THE CROSS-GRAPH TRANSFORMER

The CGT module models structural correlation across proteins and meantime achieves fine-grained learning on PPI sites. It consists two branches, i.e. ligand and receptor branches, that are co-evolved during forward inference. For each branch, two main operations named cross-graph query and multi-head attention are involed. Specifically, the Wasserstein affinity based on the optimal transport is used for cross-graph modeling in the cross-graph query. We introduce them in detail below.

**Cross-graph query.** The cross-query operation aims to measure the structual relationship between two graphs. Considering that irregular graphs don't lie in the Euclidean space, we apply the Wasserstein affinity to characterize cross-graph structural information through the optimal transport. Here, taking the ligand branch as an example, let $\mathbf{Z}_l = [\mathbf{z}_1^l, \cdots, \mathbf{z}_{n_l}^l] \in \mathbb{R}^{n_l \times d}$ represent the encoded features of the ligand graph while $\mathbf{Z}_r = [\mathbf{z}_1^r, \cdots, \mathbf{z}_{n_r}^r] \in \mathbb{R}^{n_r \times d}$ for the receptor graph, then the Wasserstein affinity of the cross ligand-receptor query, denoted as $\mathbf{A}_{l,r} \in \mathbb{R}^{n_l \times n_r}$, can be written as:

$$\mathbf{A}_{l,r} = q_l(\mathbf{Z}_l, \mathbf{Z}_r, \mathbf{\Theta}_l) = \mathbf{T}_{l,r}^\lambda \odot \mathbf{M}_{l,r}, \tag{3}$$

where $\mathbf{T}_{l,r}$ is the optimal transport matrix, $\odot$ represents elementwise production, and $\mathbf{\Theta}_l$ denotes the parameter set involved. $\mathbf{M}_{l,r} \in \mathbb{R}^{n_l \times n_r}$ is a pairwise affinity matrix with the element in the $i$-th row and $j$-th column calculated as:

$$M_{l,r}(i, j) = cosine(\mathbf{W}_Q^l \mathbf{z}_i^l, \mathbf{W}_K^r \mathbf{z}_j^r). \tag{4}$$

$\mathbf{W}_Q^l, \mathbf{W}_K^r \in \mathbf{\Theta}_l$ are learnable matrices corresponding to the ligand and receptor.

Simlar with the cross ligand-receptor query, the cross receptor-ligand query $q_r(\mathbf{Z}_r, \mathbf{Z}_l, \mathbf{\Theta}_r)$ calculates the Wasserstein affinity $\mathbf{A}_{r,l} \in \mathbb{R}^{n_r \times n_l}$ using with corresponding parameters, e.g. $\mathbf{W}_Q^r, \mathbf{W}_K^l$.

**Multi-head attention.** Based on those cross-graph query-resulted Wasserstein affinities $\mathbf{A}_{l,r}, \mathbf{A}_{r,l}$, we further derive the multi-head attetion to adaptively highlight those salient pairs of residues. For simplication, we take the ligand branch with one-head attention as an example, which can be easily extended to the multi-head case. Formally, for the ligand branch:

$$\mathbf{F}_l = f_l(\mathbf{Z}_l, \mathbf{Z}_r, \mathbf{\Phi}_l) = softmax(\mathbf{A}_{l,r})\mathbf{V}_l. \tag{5}$$

Here, $\mathbf{\Phi}_l$ denotes the parameter set involved in the multi-head attention, $softmax(\cdot)$ applies softmax operation along the rows of $\mathbf{A}_{l,r}$ and results in a $n_l \times n_r$ dimension matrix for adaptively weighting the nodes in the value graph $\mathbf{V}_l = \mathbf{Z}_r \mathbf{W}_V^l, \mathbf{V}_l \in \mathbb{R}^{n_r \times d'}$. $\mathbf{F}_l \in \mathbb{R}^{n_l \times d'}$ is the output.

The above cross-graph query and Multi-head attention operations fulfill the cross-graph tranformer with one-head attention. And the multi-head attention $\mathbf{F}_l^m, \mathbf{F}_r^m$ can be achieved by conducting parallel multi-channel transformation $f_l^m(\cdot), f_r^m(\cdot)$ based on Eqn. (3)-(5), and then aggregating all those channels. Moreover, multi-layer CGT modules can be stacked in a deep architecture with the ligand and receptor branches co-evoled, where the output features of the previous layer are concatenated with those of the current layer. The whole process is shown in Algorithm 1.

**The optimal transport.** In the learning process above, one crucial step is to learn the matrix $\mathbf{T}_{l,r}^\lambda$ in Eqn. (3) for calculating the Wasserstein affinity between two cross-graph node sets. Formally, $\mathbf{T}_{l,r}^\lambda \in \mathbb{R}_+^{n_l \times n_r}$ is the solution of an entropy-smoothed optimal transport problem:

$$\mathbf{T}_{l,r}^\lambda = argmin \, \lambda \langle \mathbf{T}_{l,r}, \mathbf{M}_{l,r} \rangle - \Omega(\mathbf{T}_{l,r}), \tag{6}$$

where $\langle A, B \rangle = tr(A^T B)$, $\Omega(\mathbf{T}_{l,r})$ is a discrete joint probability distribution calculating the entropy of $\mathbf{T}_{l,r}$. Specifically, the optimization problem in Eqn.(6) can be efficiently solved through Sinkhorn's fixed point iterations(Cuturi, 2013), and the solution can be written as:

$$\mathbf{T}_{l,r} = diag(\mathbf{u}_{l,r})\mathbf{K}_{l,r}diag(\mathbf{v}_{l,r}) = \mathbf{u}_{l,r}\mathbf{1}_{n_r}^T \odot \mathbf{K}_{l,r} \odot \mathbf{1}_{n_l}\mathbf{v}_{l,r}^T, \tag{7}$$

where $\mathbf{K}_{l,r}$ is calculated based on the distance matrix $\mathbf{M}_{l,r}$ with $\mathbf{K}_{l,r} = e^{-\lambda \mathbf{M}_{l,r}}$. In Sinkhorn iterations, $\mathbf{u}_{l,r}$ and $\mathbf{v}_{l,r}$ are kept alternately update. Taking the k-th iteration as an example, the update takes the following form:

$$\mathbf{v}_{l,r}^k = \frac{\mathbf{1}_{n_r}/n_r}{\mathbf{K}_{l,r}^T \mathbf{u}_{l,r}^{k-1} - 1}, \quad \mathbf{u}_{l,r}^k = \frac{\mathbf{1}_{n_l}/n_l}{\mathbf{K}_{l,r}\mathbf{v}_{l,r}^k}, \tag{8}$$

with $\mathbf{u}_{l,r}^0 = \mathbf{1}_{n_l}$ as an initialization.

---

**Algorithm 1:** Stacking multi-layer CGT modules.

---

**Input:** Encoded ligand and receptor graph features $\mathbf{Z}_l, \mathbf{Z}_r$; number of stacked layers $K$

1  **for** $k \leftarrow 1$ **to** $K$ **do**

     // Conducting cross-graph query according to Eqn.(3-8), then calculating multi-head attention through applying the multi-channel version of Eqn.(5) based on $\mathbf{A}_{l,r}^{(k)}, \mathbf{A}_{r,l}^{(k)}$;

2     **if** $k = 1$ **then**

3          $\mathbf{F}_l^{(k)} = f_l^m(\mathbf{Z}_l, \mathbf{Z}_r, \mathbf{\Phi}_l)$;

4          $\mathbf{F}_r^{(k)} = f_r^m(\mathbf{Z}_r, \mathbf{Z}_l, \mathbf{\Phi}_r)$;

5     **else**

6          $\mathbf{F}_l^{(k)} = f_l^m(\mathbf{O}_l^{k-1}, \mathbf{O}_r^{k-1}, \mathbf{\Phi}_l)$;

7          $\mathbf{F}_r^{(k)} = f_r^m(\mathbf{O}_r^{k-1}, \mathbf{O}_l^{k-1}, \mathbf{\Phi}_r)$;

8     **end**

     // node feature update;

9     $\mathbf{O}_l^{(k)} = f_{output}(Concatenate(\mathbf{O}_l^{(k-1)}, \mathbf{F}_l^{(k)}))$;

10    $\mathbf{O}_r^{(k)} = f_{output}(Concatenate(\mathbf{O}_r^{(k-1)}, \mathbf{F}_r^{(k)}))$;

11 **end**

**Output:** The output graph features $\mathbf{O}_l^{(k)}, \mathbf{O}_r^{(k)}$ of the $K$-layer CGT module

---

## 4 EXPERIMENT

In this section, we first introduce the used datasets and the experiment setup including the implementation details, parameter setting and the evaluation metircs. Then we compare our model with other baseline methods. Finally, we analyze our method through ablation experiments.

### 4.1 DATASET

The employed datasets are DB3 (Hwang et al., 2008), DB4 (Hwang et al., 2010), DB5 (Vreven et al., 2015), and DB5.5 [1] from the family of Docking Benchmark (DB) datasets. Each dataset is split into training and test sets. DB5.5 is the latest and largest dataset consisting of totally 271 complexes, which provides the bound state of two given proteins as well as the unbound state of each protein. The DB5 is the subset of the DB5.5 with 230 complexes, which is most widely used. The DB3 and DB4 are subsets of DB5, which are relatively small. Specially, the DB3 contains 119 complexes and DB4 has 171 complexes totally.

### 4.2 EXPERIMENT SETUP

**Data Preprossing** Considering the irregular structure of proteins, graphs are constructed to describe them by taking residues as nodes with their edges defined according to the spatial relationship. On all datasets, we extract the same features and labels by following the work in (Fout et al., 2017). Generally, the features are divided into two types, i.e. sequence features and structural features. Specially, the Position Specific Scoring Matrix(PSSM) is extracted from amino acid sequence as sequence features. For structural features, we extract Relative Accessible Surface Area(rASA), residue depth, protrusion index, hydrophobicity and half sphere amino acid composition. Finally, we combine all the aforementioned features, which result in 70-dimension features for nodes. Edge features are composed of average atomic distance and the angle between two residues (Fout et al., 2017). To define the label, a pair of amino acids are treated to be interactive if any two non-hydrogen atoms, each from one amino acid, are within $6\mathring{A}$, which is also adopted by (Fout et al., 2017; Townshend et al., 2019).

**Implementation Details** For inputs of ligand–receptor pairs, we employ graph encoders of siamese-like networks and each graph encoder is a two-layer GNN module, resulting in features of 512. For the CGT module, we stack it into a two-layer architecture, and $f_{output}$ in Algorithm 1 is set as a 1-layer GNN. The output 512-dimension features of the two-layer CGT are further passed through two layers of MLP for prediction. For a protein, the number of positive is far less than that of negative

---

[1]https://zlab.umassmed.edu/benchmark/

samples, so we employ the weighted cross-entropy loss to train our model in an end-to end manner. Specially, the weight for positive samples is 0.1 while 1 for negative ones. We train our model with stochastic gradient descent optimizer (SGD) for 200 epochs on 2080Ti GPU cards, where the learning rate is 0.1.

**Protocol** For the Docking Benchmark (DB) datasets, training sets are generally complexes from previous version dataset and test sets are those updated in the current version. For example, we split DB5.5 into a training set corresponding to DB5 and a test set composed of the complexes added in the update from DB5 to DB5.5. We use the same training set and test set as in all baseline methods (Fout et al., 2017; Townshend et al., 2019; Liu et al., 2020; Schütt et al., 2017; Duvenaud et al., 2015) for the DB5 dataset. For the DB3, DB4 and DB5.5 datasets, we use the fixed training examples and test examples shown on the website[1]. Following the previous works (Fout et al., 2017; Townshend et al., 2019; Liu et al., 2020), we evaluate the performance by median area under the receiver operating characteristic curve (MedAUC) across all complexes in the test set. The experiments are repeated 10 times and we use the average of the ten runs as the final performance. For each method we report the mean and standard deviation of MedAUC across 10 runs.

## 4.3 EXPERIMENTAL RESULTS

We compare the CPWT framework with state-of-the-arts methods, including Node Average (NA) (Fout et al., 2017), Node and Edge Average(NEA) (Fout et al., 2017), Deep Tensor Neural Networks (DTNNs) (Schütt et al., 2017), Single Weight Matrix (MFN) (Duvenaud et al., 2015), Siamese Atomic Surfacelet Network (SASNet) (Townshend et al., 2019) and High-Order Pairwise Interactions (HOPI) (Liu et al., 2020). The results are shown in Table 1. Specifically on the DB5 dataset, we compare our performance with all baseline methods reported in (Fout et al., 2017; Townshend et al., 2019; Liu et al., 2020), while the results on DB3 and DB4 datasets of the work (Liu et al., 2020) are not compared here because different protocols are used. According to the results in Table 1, our proposed CPWT method outperforms all baselines on all DB datasets. Specifically, in the most widely used DB5 dataset, our method achieves approximately 0.041 performance gain comparing with the previous work (Fout et al., 2017). It is worth noting that the recent work (Liu et al., 2020) reports relatively high performance among those compared methods on the DB5 dataset. However, it conducts additional data augmentation by incorporating in-protein pairwise interactions as positive pairs of samples. Comparing with this method, we still obtain better performance without the data augmentation. On the DB3 and DB4 datasets with less protein complexes, SASNet and several graph-based methods get poor performance. However, our method still achieve good performance. For the DB5.5 dataset, which is the largest one, our method still get the best result, with the performance gain of 0.013 comparing with NA (Fout et al., 2017).All the results above demonstrate the effectiveness of our proposed CPWT framework and the robustness against variation of dataset size.

Table 1: Comparsion with state-of-the-art-methods.

| Methods | DB3 | DB4 | DB5 | DB5.5 |
|---|---|---|---|---|
| NA (Fout et al., 2017) | 0.895 (0.015) | 0.902 (0.005) | 0.882 (0.007) | 0.931 (0.007) |
| NEA (Fout et al., 2017) | 0.881 (0.009) | 0.898 (0.005) | 0.898 (0.005) | 0.917 (0.008) |
| DTNNs (Schütt et al., 2017) | 0.853 (0.017) | 0.870 (0.015) | 0.880 (0.007) | 0.924 (0.013) |
| MFN (Duvenaud et al., 2015) | 0.875 (0.019) | 0.878 (0.009) | 0.871 (0.013) | 0.929 (0.012) |
| SASNet (Townshend et al., 2019) | 0.800 (0.012) | 0.802 (0.014) | 0.876 (0.037) | 0.910 (0.008) |
| NeiA+HOPI (Liu et al., 2020) | - | - | 0.919 (0.015) | - |
| NeiWA+HOPI (Liu et al., 2020) | - | - | 0.930 (0.016) | - |
| CPWT | 0.917 (0.001) | 0.913 (0.005) | 0.939 (0.002) | 0.944 (0.008) |

## 4.4 ABLATION STUDY

As our CPWT framework has achieved promising performance compared to the existing state-of-the-art methods, it is interesting and meaningful to make clear how the modules or parameters setting, e.g. the number of CGT layers and graph encoder layers, influence the performance of protein interaction prediction. For this purpose we conduct several additional experiments to explore our framework based on the DB5 dataset as follows.

**Influence of the CGT layer number.** To investigate the impact of CGT module on our framework, we stack multiple CGT layers and compare the performance between them. The results are shown in

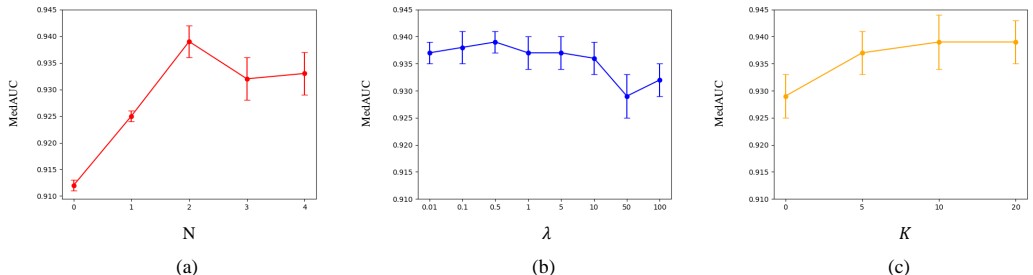

Figure 3: The evaluation of hyperparameters.(a) the number of CGT layers (denoted as N), (b) $\lambda$ in Wasserstein affinity, (c)$K$ in Wasserstein affinity)

Fig. 3 (a). When the number of CGT layers is set to 0, it means no CGT module in our framework. It can be observed that our designed CGT module effectively promote the performance, as 1.3% performance gain is obtained by using a one-layer CGT module comparing with the framework without any CGT module. Moreover,the performance varies with the number of CGT layers. And the best result is obtained when the number is set to 2.

**Effectiveness of multi-head attention module.** We compare the performance between model with or without multi-head attention. We remove multi-head attention module and use just the Wasserstein affinity (named CPWT_no_MH) to implement cross-graph function. For comprehensive evaluations, we evaluate the result in multiple setting with different GNN layers used in the graph encoder. The results are shown in Table 3. It is observed that the multi-head attention also improves the performance because the performance decreases from 93.9% to 92.3% without using the multi-head attention, which demonstrates the meaningness of conducting fine-grained learning on residues.

**Comparsion with the cross-protein reciprocal attention.** We compare the performance between the models using the Wasserstein affinity and reciprocal attention in PepNN (Abdin et al., 2022), where the cosine similarity is employed for cross-graph measurement. We replace the Wasserstein affinity with the reciprocal attention in our model, which is named as CPRA. The performaces are evaluated on all datasets and the results are shown in Table 2. As it is shown, performance improvement is obtained on all datasets, which verifies the effectiveness of our CPWT by introducing the Wasserstein affinity.

Table 2: Comparsion with reciprocal attention.

| Datasets | DB3 | DB4 | DB5 | DB5.5 |
|---|---|---|---|---|
| CPRA | 0.887 (0.003) | 0.900 (0.007) | 0.928 (0.003) | 0.935 (0.009) |
| CPWT | 0.917 (0.001) | 0.913 (0.005) | 0.939 (0.002) | 0.944 (0.008) |

**Influence of the hyperparameter in Wasserstein affinity.** In Wasserstein affinity module, there are two key hyperparameter $\lambda$ and $K$. $\lambda$ is the regularization coefficient, which controls the strength of the regularization and $K$ controls the iterative update steps. We set different values and compare them. The results are shown in Fig. 3 (c) and Fig. 3 (d). Specifically, $\lambda$ controls local information between the nodes across two graphs. For too large values of $\lambda$, the OT matrix would become little sensitive to the local correlation between two graphs and therefore would degrade the ability of capturing patterns of protein complex. Even though two parameters are varied in a pretty large range of values, the performance of the model stays above 90%.

**Influence of multiple GNN layers in graph encoder.** For comprehensive comparison, we conduct experiments with multiple numbers of GNN layers in the graph encoder, and compare the results with those baselines based on graph. The results are shown in Table 3. As shown, more GNN layers lead to the performance reduction because of the over-smoothing. Moreover, we can observe that our proposed CPWT framework can always achieve the best performance even though the number GNN layer varies, which demonstrates the effectiveness of our proposed method.

**Visualization of salient pairs of residues.** To intuitively understand the effect of our fine-grained learning, we further visualize the factor values of multi-head attention using protein complex 3V6Z as an example in Figure 4. According to Figure 4, intuitively, our proposed CGT module can endow large weights to most interactive pairs comparing with the ground truth. Specifically, compared with

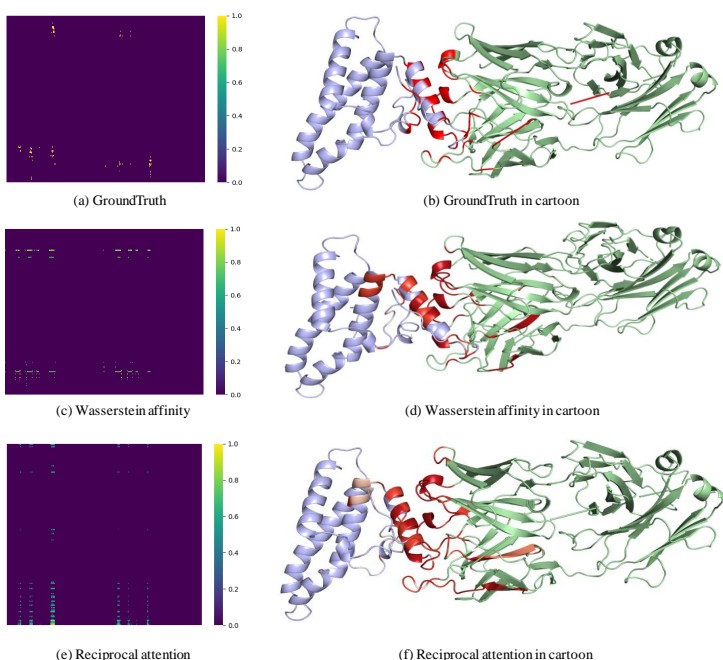

Figure 4: The visualization of the ground truth(a), (b), learned saliencies (c), (d) and reciprocal attention (e), (f) separately in the heatmap and protein cartoon model upon protein complex 3V6Z. The red parts in (b) are the binding sites of the ground truth in the complex and the red parts in (d) and (f) denotes the learned salient pairs of sites.

Table 3: Comparsion with graph-based methods on the DB5 dataset.

| GNN Layers | 1 | 2 | 3 | 4 |
|---|---|---|---|---|
| NA (Fout et al., 2017) | 0.864 (0.007) | 0.882 (0.007) | 0.891 (0.005) | 0.889 (0.005) |
| NEA (Fout et al., 2017) | 0.876 (0.005) | 0.898 (0.005) | 0.895 (0.006) | 0.889 (0.007) |
| DTNNs (Schütt et al., 2017) | 0.867 (0.007) | 0.880 (0.007) | 0.882 (0.008) | 0.873 (0.012) |
| MFN (Duvenaud et al., 2015) | 0.865 (0.007) | 0.871 (0.013) | 0.873 (0.017) | 0.869 (0.017) |
| NeiA+HOPI (Liu et al., 2020) | 0.902 (0.012) | 0.919 (0.015) | 0.921 (0.009) | 0.915 (0.009) |
| NeiWA+HOPI (Liu et al., 2020) | 0.908 (0.019) | 0.930 (0.016) | 0.924 (0.011) | 0.914 (0.013) |
| CPWT_no_MH | 0.912 (0.006) | 0.923 (0.005) | 0.921 (0.006) | 0.915 (0.005) |
| CPWT | 0.923 (0.003) | 0.939 (0.002) | 0.932 (0.002) | 0.928 (0.005) |

the reciprocal attention in (Abdin et al., 2022), our framework achieves more fine-grained learning as it has less red area but generally covers the ground truth. Hence, Table 2 and Fig. 4 jointly verify the effectiveness of our CPWT framewrok by introducing Wasserstein affinity.

## 5 CONCLUTION

In this paper, a novel CPWT framework was proposed for the PPI prediction task. Based on the hypothesis of intimate structure-function relationship, the CPWT framework focuses on cross-protein structural modeling as well as fine-grained learning on pairs of residues. Considering the irregular structure of proteins, graphs were constructed to describe them, with graph encoders then applied for robust features. Then, a core Cross-Graph Transformer (CGT) module was proposed for cross-protein structural modeling, where the cross-graph query was conducted based on Wasserstein affinities across graphs. Moreover, the multi-head attention was accordingly derived for mining fine-grained cues. For better feature learning ability, the CGT was stacked into a multi-layer structure with the ligand and receptor branches co-evolved during forward inference. Comprehensive experiments were conducted for performance evaluation, and the learned fine-grained saliencies were also visualized. All these above verified the effectiveness of our CPWT framework.

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
