# OpenReview forum: "Cross-Protein Wasserstein Transformer for Protein-Protein Interactions"
_ICLR.cc/2023/Conference — Submitted to ICLR 2023_

### Official Review · Reviewer_sndu · 2022-10-19

**Confidence:** 4
**Correctness:** 2
**Technical Novelty And Significance:** 2
**Empirical Novelty And Significance:** 3
**Recommendation:** 3

**Clarity, Quality, Novelty And Reproducibility:**

Method description seems incomplete. making reproducibility very difficult.
Figure 2: how the authors do their "sampling strategy"?
Figure 2: How authors do their "combined representation"?
What is the loss function? Binding affinity or "bind or no-bind" classification?
How is the data prepared and how the labels are derived?
More description is needed for readers who are not familiar with the data sets this paper use.
What is the data balance? The author use AUC as a measure, need to justify that AUC is a good measure for this work.

**Strength And Weaknesses:**

Key new ideas are (1) cross transformer (2) Use of Wassertien distance.
Very good that the authors repeat their experiments 10 times and generate error bars.

The method description seems incomplete. Elaborations about this is given in the next section of the review

**Summary Of The Paper:**

The paper propose a cross attention mechanism for classifying protein-protein interactions. The hypothesis is that information from both proteins get mixed during encoding, leading to better predictions. Paper also propose using the Wasserstein distance measure while processing the attention values.

**Summary Of The Review:**

I recommend that the authors revise the whole of the method section, keeping in mind reproducibility of this paper. Then resubmit to another conference or journal.

Experiments to show the effects between "cross attention" and "no cross attention" is needed. Readers may not be convinced that crossing has a significant impact on the prediction until they see good experiments. Keeping everything else unchanged, train two transformers. one with crossing of features (in this paper) and the other transformer without crossing of features.

---

### Official Review · Reviewer_1voW · 2022-10-24

**Confidence:** 4
**Correctness:** 2
**Technical Novelty And Significance:** 1
**Empirical Novelty And Significance:** 1
**Recommendation:** 3

**Clarity, Quality, Novelty And Reproducibility:**

On clarity:
While the paper is relatively easy to follow thanks to the simplicity of the proposed approach, unfortunately there are frequent grammatical errors making the manuscript hard to read. Moreover, the problem statement and prediction targets should be described more explicitly for the paper to be self-contained.

**Details Of Ethics Concerns:**

NA.

**Strength And Weaknesses:**

Strengths:
+ Despite the recent breakthroughs in protein structure prediction heralded by AlphaFold2, accurately predicting the interface of protein-protein interactions remains an open problem, making the paper's topic very relevant.

Weaknesses:
+ Experimental results for the most competitive baselines (NeiA+HOPI and NeiWA+HOPE) are lacking for three out of four datasets.
+ The experiments do not provide compelling evidence that the proposed approach outperforms the baselines by a significant margin.
+ For an application-focused paper, the experimental results lack breadth in terms of the datasets and metrics under consideration.
+ The paper lacks a significant methodological contribution.

**Summary Of The Paper:**

This paper proposes a method for protein interface prediction. In a nutshell, given two proteins (a ligand and a receptor) whose 3D structures are assumed known, the aim is to identify contacts between residues in the receptor and residues in the ligand.

To this end, the authors propose a simple model that:
1. Uses GNN layers to independently compute per-residue embeddings for the ligand and the receptor.
2. Refines these embeddings by applying (possibly several) layers of doubly stochastic cross-attention between the ligand and receptor embeddings.
3. Uses the resulting refined embeddings to predict residue-residue interactions.

The authors evaluate the resulting approach on the Docking Benchmark (DB) datasets.


**Summary Of The Review:**

As it stands, unfortunately I do not believe the manuscript to have a sufficiently significant contribution, neither methodological nor application-related, to warrant publication.

From a methodological perspective, the proposed model follows a traditional dual GNN encoder architecture with cross-attention fusion layers. Perhaps the most innovative design choice is the use of doubly stochastic cross-attention by means of the Sinkhorn algorithm in these cross-attention layers, but doubly stochastic attention is also not novel per se (see e.g. [1]).

From an application-centric perspective, the reported performance improvements are within the margin of error for the most competitive baselines (e.g. NeiWA+HOPI) whenever these are shown (only one out of four datasets), making it hard to judge the significant of the results. Moreover, in my opinion, the experimental setup itself is too limited for an application-centric manuscript. As potential directions for improvement in this regard, I would suggest all of the following:

1. Reporting NeiA+HOPI and NeiWA+HOPI performance values for all datasets.
2. Evaluating on larger, newer benchmarks such as DIPS [2] and DIPS-Plus [3].
3. Including more recent baselines in the comparison.

References:
[1] Sander, Michael E., et al. "Sinkformers: Transformers with doubly stochastic attention." International Conference on Artificial Intelligence and Statistics. PMLR, 2022.
[2] Savidor, Alon, et al. "Database-independent protein sequencing (DiPS) enables full-length de novo protein and antibody sequence determination." Molecular & Cellular Proteomics 16.6 (2017): 1151-1161.
[3] Morehead, Alex, et al. "DIPS-Plus: The Enhanced Database of Interacting Protein Structures for Interface Prediction." arXiv preprint arXiv:2106.04362 (2021).

---

### Official Review · Reviewer_jcwb · 2022-10-25

**Confidence:** 4
**Correctness:** 4
**Technical Novelty And Significance:** 2
**Empirical Novelty And Significance:** Not applicable
**Recommendation:** 5

**Clarity, Quality, Novelty And Reproducibility:**

The quality and clarity are good but the originality is only marginally significant or novel, and missed code links to reproduce experiments.

**Strength And Weaknesses:**

##########################################################################

Pros:

- This paper proposes a new Cross-Protein Wasserstein Transformer framework to promote the PPI prediction from the perspective of sophisticated cross- protein structural modeling based on Wasserstein affinities.
- This paper also proposes a novel CGT module in which the multi-head attention is derived to mine fine-grained cues of PPI sites. Moreover, this module can be stacked into a deep architecture with two branches co-evolved, which are powerful in cross-protein structural expression with sophisticated fine-grained learning.
- This paper reports the state-of-the-art performance on multiple PPI datasets with comprehensive experiments.
##########################################################################

Cons:

- The core idea of this paper is CPWT framework and CGT module, however, although the authors achieved SOTA performance on multiple PPI datasets with comprehensive experiments, seems CGT is just combined multi-head attention and Wasserstein affinity, if lacking enough theory support or empirical insight such combination novelty is limited.
- The authors claim "To learn robust features from these graphs, graph encoders such as GNNs are then applied." Why? What GNNs have to do with robustness?
- In the experiment section, all experiments are based on Docking Benchmark (DB) datasets, is there any other dataset to support the hypothesis?
- Missing code link to reproduce experiments.

**Summary Of The Paper:**

This paper proposes a novel deep learning framework named Cross-Protein Wasserstein Transformer (CPWT) to predict PPI sites through fine-grained cross-graph structural modeling, which promotes the PPI prediction from the perspective of sophisticated cross-protein structural modeling based on Wasserstein affinities. Then, a core Cross-Graph Transformer (CGT) module of two branches (e.g. ligand and receptor branches) is proposed for cross-protein structural modeling. Specifically, in this module, Wasserstein affinity across graphs is calculated through cross-graph query (i.e. ligand (query) - receptor (key) or the converse), based on which the multi-head attention is derived to adaptively mine fine-grained cues of PPI sites. By stacking CGT modules, the two branches in CGT are co-evolved in a deep architecture during forwarding inference, hence being powerful and advantageous in cross-protein structural representation and fine-grained learning. The experimental results show the effectiveness of our CPWT framework by conducting comprehensive experiments on multiple PPI datasets, and further visualize the learned fine-grained saliencies for intuitive understanding.

**Summary Of The Review:**

Considering the above pros and cons, my recommendation of the paper is marginally below the acceptance threshold.

---

### Official Review · Reviewer_HBkA · 2022-10-31

**Confidence:** 4
**Correctness:** 3
**Technical Novelty And Significance:** 2
**Empirical Novelty And Significance:** 2
**Recommendation:** 3

**Clarity, Quality, Novelty And Reproducibility:**

The paper is not entirely clear and raises quite a few questions:

- what is meant by irregular graphs when referring to protein graphs?
- why should the weights $W_Q$ and $W_K$ be different for the ligand and receptor?
- what is the formal motivation / rationale for using Wasserstein affinity when doing cross attention?
- what is the computational burden of the model with Wasserstein affinity and how does it compare with the chosen baselines?
- how is the final site prediction done? The method section ends by describing the cross-attention (second) module and has no description of the third (last) detection module.


**Strength And Weaknesses:**

- The ablation studies with regards to the importance of different components are informative.

- Given the recent advances in protein-complex folding, what is the benefit of PPI site prediction?
- The results should have been compared with the recent methods that do structure prediction for a pair or a complex of proteins.



**Summary Of The Paper:**

The paper proposes Cross-Protein Wasserstein Transformer (CWPT) for protein-protein interaction site detection. For this it proposes an elaborate architecture, composed of three main parts: (i) A graph network as the encoder of the individual proteins, (ii) a (cross-graph) transformer with cross-attention (from one protein to another) using a Wasserstein affinity, and (iii) a final module to operate on the cross-protein embeddings to detect the interaction sites.

**Summary Of The Review:**

The work does not have a technical machine learning contribution and hence should be refereed as an application paper. Given that there are quite a few recent works on protein complexes, it is unclear what application benefits the results of this work bring about. Furthermore, the presentation requires important clarifications. These together suggest that the paper may not be ready for publication.

---

### Decision · Program_Chairs · 2023-01-20

**Decision:**

Reject

**Justification For Why Not Higher Score:**

See justification above.

**Justification For Why Not Lower Score:**

NA

**Metareview: Summary, Strengths And Weaknesses:**

This paper proposes the Cross-Protein Wasserstein Transformer, a neural architectures that seek to extract features from 2 proteins (ligand and receptor) to predict where proteins will bind. The architecture includes GNNs and cross-attention mechanisms with a balanced attention (using Sinkhorn). The paper received overall negative reviews, and the authors did not provide a rebuttal, hence my decision to reject. I do regret however this, because the paper looks interesting and might become a worthy addition to the literature. I would like to encourage the authors to take into account the feedback and work on a resubmission, perhaps to a comp bio venue. More care should also be taken to explain more intuitively the architecture choices. Finally, the paper has a few unusual typos (Comparsion, conclution, etc...) that suggest it was prepared in haste.